# Mechanical adaptability of artificial muscles from nanoscale molecular action

Federico Lancia [1], Alexander Ryabchun [1], Anne-Déborah Nguindjel[1], Supaporn Kwangmettatam[1] & Nathalie Katsonis [1]*

The motion of artificial molecular machines has been amplified into the shape transformation of polymer materials that have been compared to muscles, where mechanically active molecules work together to produce a contraction. In spite of this progress, harnessing cooperative molecular motion remains a challenge in this field. Here, we show how the light-induced action of artificial molecular switches modifies not only the shape but also, simultaneously, the stiffness of soft materials. The heterogeneous design of these materials features inclusions of free liquid crystal in a liquid crystal polymer network. When the magnitude of the intrinsic interfacial tension is modified by the action of the switches, photo-stiffening is observed, in analogy with the mechanical response of activated muscle fibers, and in contrast to melting mechanisms reported so far. Mechanoadaptive materials that are capable of active tuning of rigidity will likely contribute to a bottom-up approach towards human-friendly and soft robotics.

[1] Bio-inspired and Smart Materials, MESA+ Institute for Nanotechnology, University of Twente, PO Box 207, Enschede 7500 AE, The Netherlands. *email: n.h.katsonis@utwente.nl

Recent developments in synthetic organic chemistry have supported the design and synthesis of molecular motors[1,2], rotors[3,4], switches[5–7], and other dynamic molecules that display mechanically relevant and sophisticated motion in response to external stimuli[8–11]. Strategies to make these molecules work together, in space and time, have allowed amplifying their nanoscale operation modes into purposeful shape transformation at the macroscopic level[12–18]. Notably, successful approaches have involved liquid crystal systems, in which long-range order couples to molecular motion efficiently. Examples range from chiral liquid crystals[19,20] to liquid crystal elastomers[21] and other liquid crystal networks[22], and liquid crystal networks incorporating light-responsive molecules have been developed to mimic cilia movement[23], tendrils[12,24], opening of seedpods[25], flytraps[26], and continuous wave propagation[27]. These macroscopic shape transformations, as complex as they are, have not been combined with a stimuli-induced enhancement of stiffness so far[28], which limits, e.g., the interactions that they can establish with unpredictable surroundings. Instead, illuminating polymer systems that incorporate photo-switches covalently irremediably leads to softening by ways of fluidization[29,30].

Muscle contraction is an archeypal example for cooperative operation of molecular machines, where the operation of myosin motor proteins is amplified over increasing length scales to generate work, with anisotropy and helical structuration of the actin filaments being key features of the mechanism for muscle contraction. Notably, muscle fibers display nonlinear strain-stiffening, which protects them against rupture[31]. Upon stimulation, their stiffness increases transiently. Recent work has endeavored to program stimuli-induced stiffness increase in man-made artificial materials[32–35]. These mechanically responsive systems remain, however, inherently passive and being isotropic materials, this pre-programmed change in stiffness is not combined with contraction.

Here, we show actuating materials that also display complex mechanical adaptability by design, including both light-induced stiffening and light-induced softening, as well as nonlinear responses to stress. Our strategy involves incorporating phase heterogeneity in liquid crystal polymer networks, by adding a free liquid crystal that forms inclusions in the network. Over a given threshold for the amount of interfacial area, illumination decreases the miscibility between the liquid crystal and the polymer network, because the polymer becomes less ordered and more polar and this decrease of miscibility leads to an increase in the interfacial contribution to mechanical properties, and eventually to photo-stiffening. These mechanically adaptive materials can convert the work produced by molecular switches efficiently, by generating stresses in the MPa order. Moreover, they display light-responsive enhancement of their nonlinear response to stress, which is a salient characteristic of myosin-activated muscle fibers.

## Results

### Bio-inspired design for mechanical adaptability

The design is based on photo-responsive liquid crystal networks that are formed in the presence of increasing proportions of liquid crystal, which does not take part into the covalent network. Upon cross-linking, the polymer network phase separates spontaneously from the liquid crystal, and this phase separation yields a large interfacial area. When the resulting material is illuminated with ultraviolet light, activation of the cross-linking molecular switches modifies both the polarity and the morphology of the polymer network (Fig. 1), thereby diminishing the miscibility of the two components, and we show that this molecular event has a major consequences on the stiffness of the material.

This design thus combines long-range order and the properties of interface-dominated materials with the stimuli-responsive properties of azobenzene switches. Azobenzene switches convert from a rod-like *trans* form to a bent *cis* form under irradiation with UV light, and this transformation is reversible once the light is turned off. The switch was incorporated in a liquid crystal polymer network as a cross-linker. The liquid crystal network was formed in situ, in a nematic liquid crystal that is composed by cyanobiphenyls, and is known to be partially miscible with the liquid crystal polymer network (its composition is shown in Supplementary Fig. 1).

Overall, the materials show some analogy with mutable collagenous tissues, in which the interplay between collagen fibers and a viscoelastic matrix results in a nonlinear mechanical response that is combined with stiffening under mechanical stress[36,37]. Approaches combining soft polymers with rigid cellulose nanofibers have yielded chemoresponsive materials that soften upon exposure to water[38], or stiffen upon heating[39]. Temperature-induced stiffening has also been engineered in interpenetrated networks of temperature-responsive gels, with the necessity to heat and the absence of shape transformation[40].

The proportion of liquid crystal in the material was allowed to vary, with $R$ being the molar ratio between the free liquid crystal and the azobenzene cross-linker (see Supplementary Table 1 for compositions). The morphology of these materials was characterized by atomic force microscopy, in the presence of the liquid crystal, and also after the free liquid crystal was washed out (Fig. 2). After washing, the morphology was typical of a liquid crystal network, and revealed distinct differences in structure and porosity between the upper side and the bottom side of the film. This difference in pore sizes originates from asymmetric irradiation conditions that create a gradient in cross-linking density through the thickness of the film. We previously exploited such a gradient to control the shape-shifting of polymer springs[12]. After washing the free liquid crystal away, the characteristic stretching band of the nitrile group at ~ 2200 $cm^{-1}$ disappeared from the infrared spectrum, which confirmed phase heterogeneity (Supplementary Fig. 2). Such heterogeneity corresponds to nanoscale porosity, whereas microscale pores are formed for proportions of free liquid crystal typically exceeding 50 wt% (Supplementary Fig. 3). Besides, scanning electron microscopy of cross-sections shows a clear difference in morphology between the fully polymerized network and the phase-separated materials (Supplementary Fig. 4). We note that it is not possible to prepare a similar material where the pores of the network would be filled with an isotropic liquid, because the presence of an isotropic liquid in large quantities compromises the liquid crystallinity of the polymer network.

### Active change of stiffness in response to light

The stiffness was investigated in thin and flat ribbons of materials, prepared by cutting planar films in a direction parallel to the liquid crystal director. The stiffness of the ribbons was measured by stretching them continuously, either in ambient conditions, or under irradiation with ultraviolet light (Fig. 3a shows a schematic of the experimental setup, the light intensity was fixed at 17 mW/cm²). The relative extension of the composite, e.g., the strain, was recorded against the stress applied by the pulling forces, to yield a stress–strain curve (Fig. 3). The slope of this curve equals the Young modulus, which is a measure of how much materials resist to deformation—with materials becoming softer when the Young modulus decreases, and stiffer when it increases. In our experiments, the ribbons are always stretched along the liquid crystal director and any substantial contribution from soft elasticity is thus excluded[41].

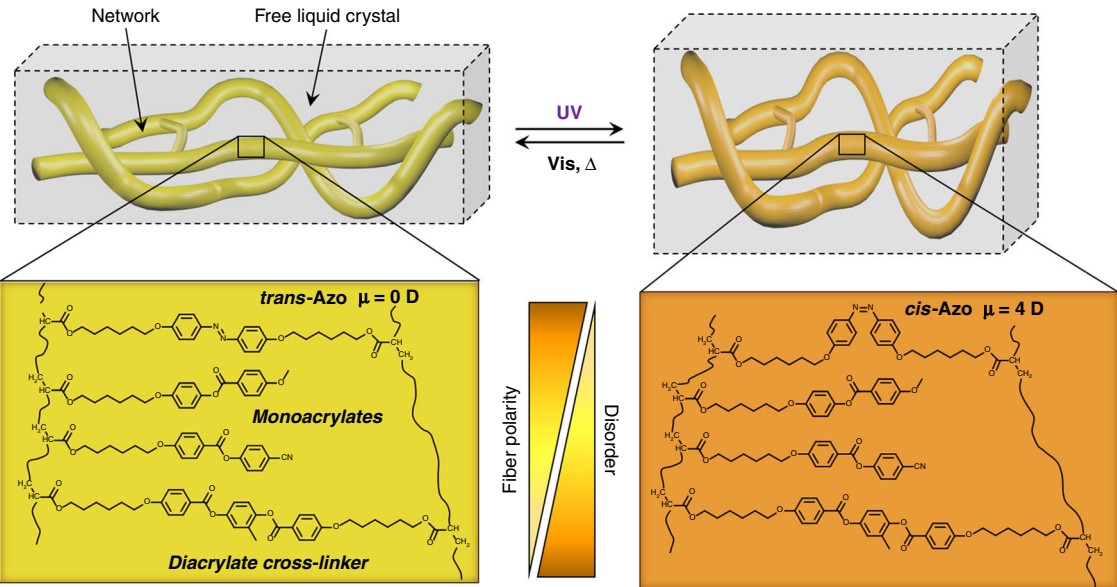

**Fig. 1** Design and composition of mechanically adaptive polymers. The zoom shows the molecular components of the network. The network and the free fraction of liquid crystal phase separate. The photo-induced isomerization of the azobenzene induces disorder and increases the polarity of the fibers, and therefore isomerization impacts the orientation (anchoring) of the free liquid crystal molecules, and their miscibility to the network

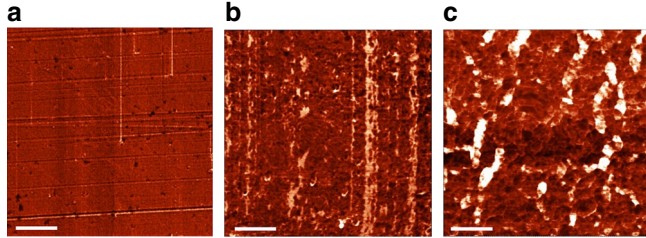

**Fig. 2** Morphological investigations. Atomic force microscopy images of the mechanically adaptive polymer composites (phase mode). **a** Morphology of the original material. **b, c** After the free liquid crystal is washed away with acetonitrile, the polymer network is revealed. The morphology of the material differs between the top side **b** and on the bottom side of the thin film **c**. Scale bars are 1 μm

In ex situ experiments, two different ribbons were used to quantify stiffness; one in the absence of any irradiation, and another one was measured only after all molecular switches had reached photo-stationary state (Fig. 3b, e). Ex situ measurements allow the modulus to be determined precisely, with no artefacts arising from pre-stretching. In situ experiments were carried out with a ribbon that was extended, first under ambient light conditions, and next under UV illumination (Fig. 3c, f).

In the absence of irradiation, stress–strain curves show a linear response to pulling, indicating that the deformation is completely elastic and therefore reversible, i.e., no plastic deformation is observed until the ribbons break (Fig. 3b, e).

Upon irradiation with UV light, the materials show different mechanical properties, depending on the proportion of liquid crystal they contain. With a low proportion of free liquid crystal, photo-induced softening is consistently and exclusively observed for both in situ and ex situ illumination conditions (Fig. 3b, c). Such a photo-induced softening is in line with earlier reports on the behavior of liquid crystal and amorphous azobenzene-containing polymers, in the absence of any phase heterogeneity. Shimamura et al.[29] have reported that irradiation of cross-linked azobenzene-containing liquid-crystalline polymer films leads to a decrease of the Young modulus, and Kumar et al.[30] have also documented the photo-fluidization of similar materials. Light-induced decrease of modulus has been reported in all mechanically responsive azobenzene polymers reported so far[42,43]. In these homogeneous materials, the rod-like shape of the trans-azobenzene is compatible with the liquid crystal order, whereas the crescent shaped cis-azobenzene that is formed by light disrupts the liquid crystal organization. As the trans-to-cis photo-isomerization of the switches disrupts the organization of the polymer, the network undergoes light-induced softening by fluidization.

By contrast, and rather counter-intuitively, increasing the percentage of free liquid crystal eventually makes the material stiffen in response to light, and both ex situ and in situ photo-tensile experiments evidence a large photo-induced increase of the Young modulus (Fig. 3e, f).

The swelling degree is a critical factor in the response of these mechanoadaptive materials. In the absence of light, the Young modulus decreases linearly when the proportion of free liquid crystal increases, from an average 220 MPa to 60 MPa (Fig. 3g). Such a softening is not surprising for small molecules that are known to behave as plasticizers of polymer materials[44]. Under illumination, the modulus keeps decreasing with increasing proportion of free liquid crystal, however this decrease becomes less pronounced. For compositions beyond $R \approx 3.9$ large photo-stiffening responses are observed (Fig. 3f).

The photo-induced variation of modulus $\Delta E\% = (E_{PSS} - E_0)/E_0 \times 100$ (where $E_{PSS}$ is the Young modulus at the photo-stationary state and $E_0$ is the Young modulus in ambient conditions) was plotted as a function of the swelling degree. Arbitrarily, we attribute negative values to softening, and positive values to stiffening. We observe that the stiffness varies more with larger proportions of free liquid crystal (Fig. 3g). The materials exhibit a large mechanical adaptability, softening up to 40% and stiffening up to 150% under illumination. A comparable magnitude in softening or stiffening behavior has been observed in temperature-responsive hydrogels, by collapse of polyethylene glycol polymer chains[39]. In hydrogels doped with molecular motors, light induced an increase in the shear modulus of the material, that relates to viscosity[45]. However, stimuli-induced increase of the Young Modulus was not reported and, typically,

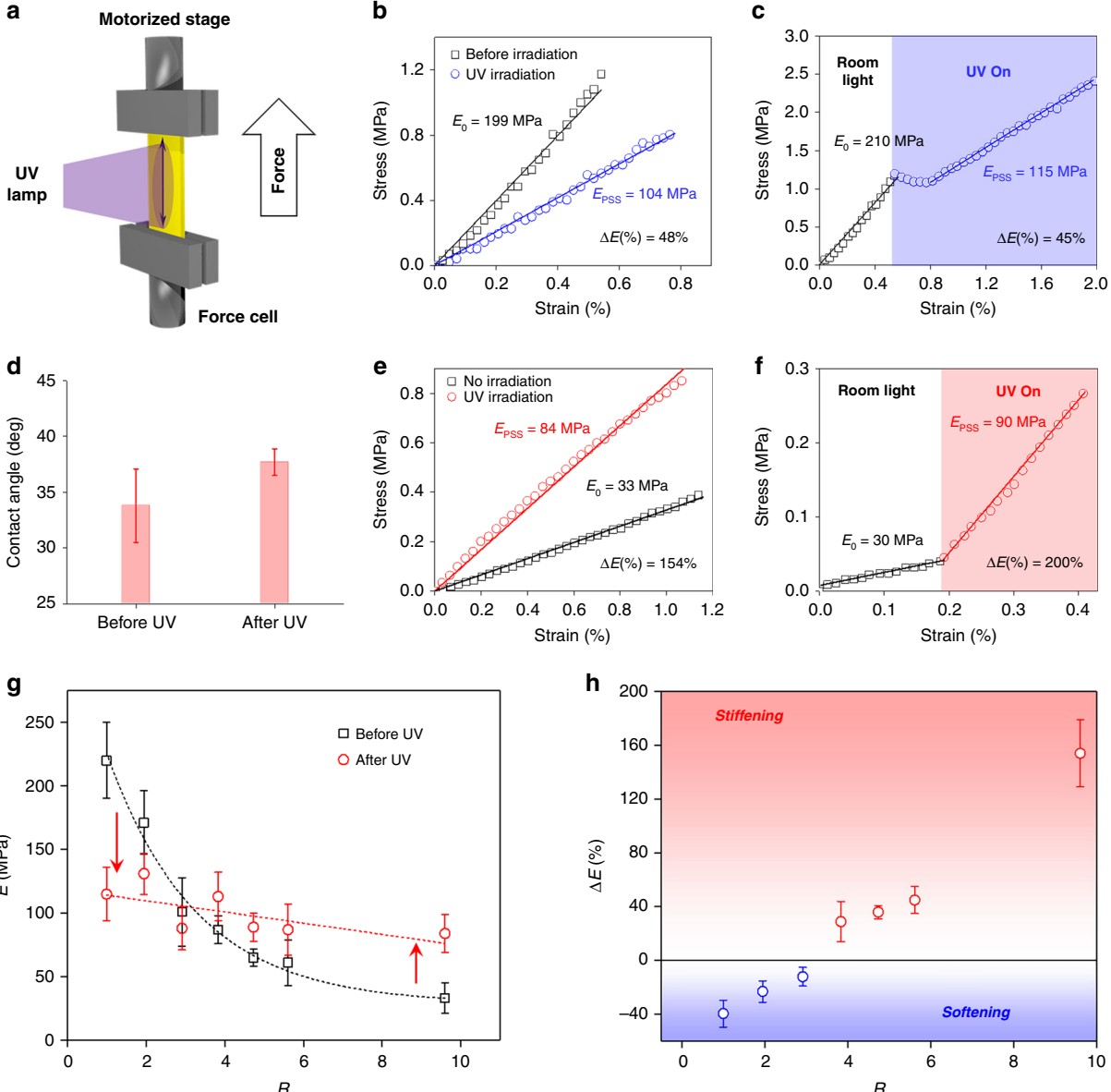

**Fig. 3** Active modulation of stiffness in response to light. **a** Photo-tensile experiments on flat ribbons. The polymer is stretched parallel to the molecular alignment, as indicated by a black double-headed arrow. The stiffness of the material adapts to illumination. **b** Ex situ stress–strain curves for a $R = 1$ material, in room-light condition (black squares) and under irradiation (blue circles). We define $R$ as the molar ratio between the free liquid crystal and the molecular photo-switch. **c** In situ stress–strain curve for material with $R = 1$. The film is initially strained in ambient conditions (black squares) and then exposed to UV light (blue circles). When the light is switched on, the material requires time to adjust to the new illumination conditions, which is visible in in situ tensile measurements. **d** Contact angle of free liquid crystal on a fully polymerized, unswollen polymer network, before and after activation with UV light. The contact angle increases upon light activation. **e** Ex situ stress–strain curves for material with $R = 9.6$ in room-light condition (black squares) and under irradiation (red circles). **f** In situ stress–strain curves: the material is first strained (black squares) and then exposed to UV light (red circles). **g** Average Young modulus before (black squares) and after irradiation (red circles), for increasing proportions of free liquid crystal in the material. Error bars correspond to standard deviation. **h** Photo-induced variation of the Young modulus, for increasing proportions of free liquid crystal in the material. Error bars correspond to standard deviation

the modulus of hydrogels remains modest, which constitutes a limitation in the production of macroscopic work.

Reversibility in the photo-stiffening behavior is shown in Supplementary Fig. 5, where irradiation of a flat ribbon with visible light converts the azobenzene back to its *trans* form, and the Young modulus consequently returns back to its original value. Reversibility in mechanical adaptation indicates that no degradation occurs as a result of illumination with UV light.

We propose that the photo-stiffening originates in a decrease in solubility of the free liquid crystal in the polymer network. This

light-enhanced incompatibility creates elasto-capillary forces that oppose the formation of interfacial area and thus tends to resist against deformation—which is observed as a stiffening. This mechanistic understanding is supported by contact angle experiments performed on full polymer networks supporting liquid crystal droplets: the contact angle increases under illumination, indicating that the interfacial tension increases with light (Fig. 3d). Recent work on ultrasoft, silicone-based materials has also shown that microscopic liquid inclusions can increase the stiffness of soft solids when interfacial tension opposes elastic

deformation—provided that these inclusions are sufficiently small[46]. Here, we show that light enhances the contribution of the interfacial tension that is intrinsic to the material (Fig. 3d), because compatibility between the free liquid crystal and the network decreases when the *cis*-rich network becomes too polar and disordered to be compatible with the free liquid crystal. Recent literature also unequivocally demonstrates that nonreactive liquid crystals can swell liquid crystal networks only up to a certain threshold, and this partial miscibility decreases markedly once the azobenzene has switched to the *cis* form[47,48]. Ryabchun et al.[47] have shown how a liquid crystal gel with a low polymer content expels a free liquid crystal doped with azobenzenes upon irradiation with light. Broer et al. have reported that azobenzene-activated smectic gels behave as photo-sponges, because of a photo-induced miscibility with the liquid crystal solvent. For concentrations such as reported here, where the free liquid crystal fraction remains below 50% of the total weight, expulsion of the free fraction does not occur[48]. Essentially, mechanical adaptability originates in the operation of the molecular switches, provided that there is enough free liquid crystal to create a critical amount of interface, over which elasto-capillary effects become significant.

**Effect of illumination intensity on the actuation stress**. Liquid crystal networks incorporating molecular switches change shape in response to light[12,24,25]. These shape-shifting properties are preserved in mechanoadaptive materials. In parallel to stiffness changes, shape changes allow this material to convert light into mechanical work[12]. We have investigated how much stress mechanically active materials can impart with increasing illumination conditions (Fig. 4a). A flat ribbon was clamped between a force sensor and a motorized stage, and prestrained at +0.1% its original length. When keeping the elongation constant, the force sensor records the stress that is applied by the contraction. Two regimes can be distinguished, depending on the illumination conditions (Fig. 4a). At low intensities (17 mW/cm²), the stress generation follows the kinetics of the molecular switch, and eventually reaches a plateau when the photo-stationary state is established. In this regime, the stress is exclusively generated by the operation of the molecular switch. With stronger illumination conditions (above 38 mW/cm²) both the generation and the release of stress are instantaneous, which is typical for temperature-driven actuators that quickly dissipate heat after exposure. An infrared camera confirms that, under stronger illumination conditions than the ones used here, the ribbon reaches up to 85 °C (Fig. 4b), which is well above the glass transition temperature (Supplementary Fig. 6). The operation of the molecular switch and the increase in temperature add up to decrease the order of the material, and these two effects amplify the shape changes. Once the light is switched off, the temperature drops down and thus the stress generated by the material returns to that of the photo-stationary state. As the *cis* form of the molecular switch is stable over dozens of hours, the back-isomerization from the *cis*-azobenzene to the *trans*-azobenzene does not occur in ambient irradiation conditions over the time scales of the experiment (Fig. 4a). However, upon illumination with visible light, the relaxation is accelerated, the stress is released, and this constitutes further proof of that the mechanical response of these materials is reversible. Overall, photo-stiffening materials impart higher photogenerated stress than photo-softening ones and we show how illumination conditions can be used to tune the force exerted by these materials (Fig. 4c).

In addition to modulating the work that can be produced by shape transformation (e.g., bending, Supplementary Fig. 7), light intensity also modifies the magnitude of the stiffness changes in the material. Under harsh illumination conditions (~380 mW/cm²), photo-softening materials showed enhanced softening that was likely caused by fluidization (Fig. 4d). In contrast, even under harsh illumination conditions, photo-stiffening ribbons retained a clear stiffening response, albeit less pronounced, as a result of photo-stiffening competing with temperature-induced softening by fluidization (Fig. 4e). Besides these intensity-dependent experiments, all the tensile experiments we describe hereafter and above were performed under mild illumination conditions (17 mW/cm²). In these conditions there is no temperature increase, which means that dynamic stiffness is determined by the action of the molecular switches alone.

**Muscle-like stiffening of mechanoadaptive springs**. Tunable stiffness, that includes not only softening but also stiffening, is a quintessential requirement for the development of artificial muscles. At rest, passive muscle fibers stretch in a nonlinear fashion (Fig. 5a). Stiffening in response to elongation is an in-built mechanical property that protects the fibers from breaking[31]. In activated muscle fibers, the steeper slope of the force curve highlights that this nonlinear response spans a range of increased stiffnesses, which is a prerequisite to muscle performance. Such a behavior can be encoded intrinsically in mechanoadaptive materials, to not only mimic the static mechanical responses of skeletal muscles but also the active adaptation processes they undergo while at work.

Springs were prepared by adding a small amount of chiral dopant and confining the material in a glass cell that promotes a twist across the thickness of the sample (Fig. 5b). After polymerization, ribbons cut out from the film undergo curling and twisting, as a result of the chiral organization of the molecules combined with a gradient in cross-linking density[12,24]. These springs wind or unwind in response to light, depending on their geometry. They lift >10 times their weight, thereby performing measurable work (Supplementary Fig. 8). When stretched, the diameter and pitch of the springs are modified (Fig. 5c), and therefore they display nonlinear stiffening in response to strain (Fig. 5d), like all soft springs do[49]. The spring elongation shows negligible hysteresis upon cycling because most of the energy goes into geometrical changes (Fig. 5d). Only once the springs are unwound are the pulling forces applied to the covalent network itself. The transition from a non-stressed to a stretched network is responsible for the typical nonlinear response.

Combining the nonlinear behavior of the springs with active stiffness allows encoding muscle-like mechanical adaptability (Fig. 5). When a spring is stretched in room-light condition, a nonlinear mechanical response is observed and under irradiation with light, this curve shifts towards lower strains, i.e., the spring stiffens. Reversibly, upon illumination with visible light, the spring softens back until reaching a stiffness that is similar to its original value (Fig. 5e). The opposite behavior is observed for springs made of photo-softening materials (Supplementary Fig. 9).

## Discussion

The sophistication of synthetic organic chemistry has led to the design and realization of artificial molecular machines and of biomimetic polymers. Here, we add mechanical adaptability to light-responsive polymer materials by integrating molecular switches into interface-dominated polymer networks. In combination with the mechanical action of the molecular switches, the anisotropy of the material mediates shape-shifting, whereas the dynamic phase heterogeneity provides mechanical adaptability in the form of nonlinear stiffening. Using a molecularly active source of stress thus mediates light-induced stiffening, by rational

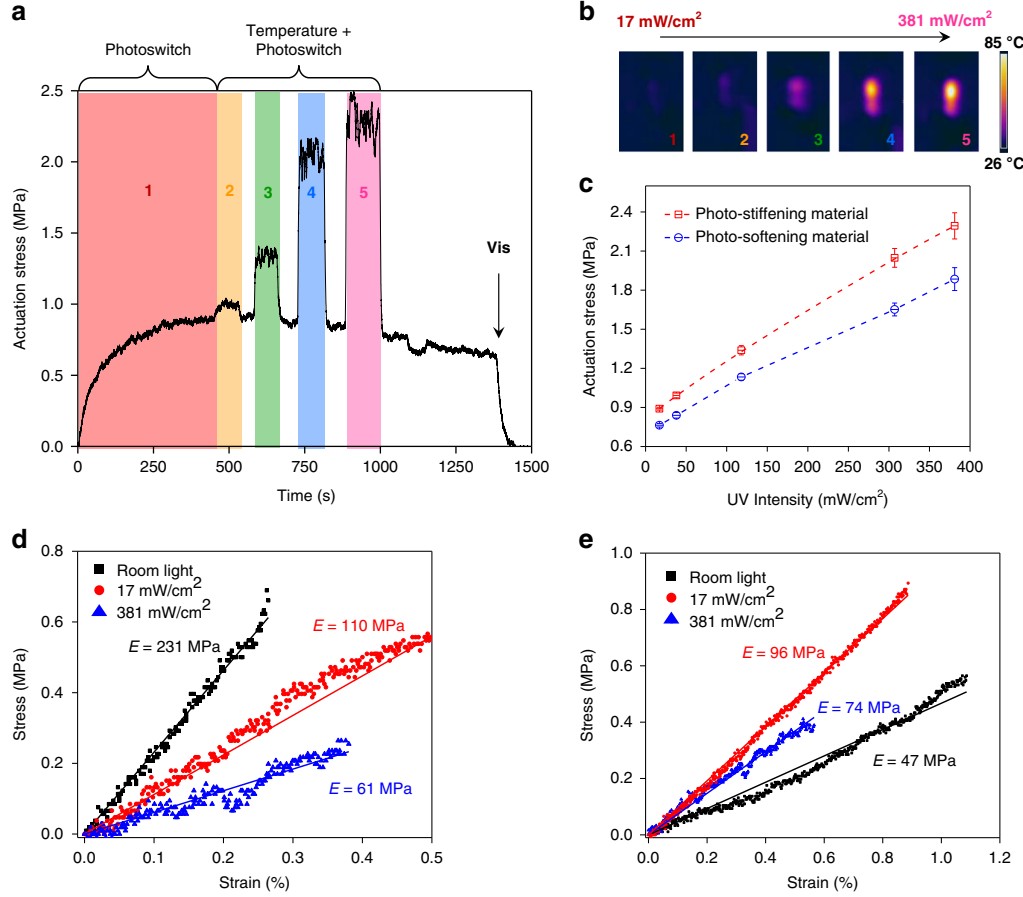

**Fig. 4** Influence of light intensity on the possibility to produce work and on the magnitude of the stiffness increase. **a** Actuation stress imposed by the photo-stiffening polymer ($R = 4.7$), for increasing intensities of light ($\lambda = 365$ nm): 1 (17 mW/cm$^2$), 2 (38 mW/cm$^2$), 3 (118 mW/cm$^2$), 4 (307 mW/cm$^2$), and 5 (381 mW/cm$^2$). This curve does not relate to light-induced stiffness changes of the material, instead it is related to its light-induced shape changes, and because of these shape changes the material exerts a force on the cantilever, i.e., in other words this experiments shows the potential for this material to convert light into mechanical work. At low intensities, the response of the polymer, i.e., its tendency to change shape, is regulated by the molecular switch. At higher intensities (2–5), the temperature intervenes. **b** Infrared images of a photo-stiffening material under different intensities of light irradiation. **c** Stress exerted by photo-stiffening materials (red squares) and photo-softening materials (blue circles, $R = 1$) as a function of light intensity. Error bars correspond to standard deviation. **d** Tensile test on a photo-softening ribbon, reported for three different illumination conditions. Here, the tensile stress is the force causing the deformation of the material, and the strain is the response of the material by ways of elongation. In contrast to the experiment reported in Fig. 4a, here the response of the material relates to changes in the Young Modulus (i.e., tensile elasticity). **e** Tensile test on a photo-stiffening ribbon, for three different illumination conditions. Photo-stiffening occurs even under harsh illumination, in spite of the increase of temperature that competes with the stiffening

engineering of intrinsic interfacial tension. Whilst being light, these actuating materials can lift more than ten times their weight. They react fast, reversibly, display large shape changes in response to illumination, and they combine a broad range of actuation modes with active stiffening. Overall, integrating artificial molecular switches in anisotropic soft matter shows potential to create mechanoresponsive materials that are able to combine fast and complex deformation modes with some of the complex mutable properties of skeletal muscles and the mechanical adaptability of biological tissues.

## Methods

**Reagents and chemicals**. Monoacrylate 4-methoxybenzoic acid 4-(6-acyloyloxy-hexyloxy)phenyl ester, 97% (Synthon Chemicals), monoacrylate 4[4[6-acyloxyhex-1-yl)oxyphenyl]carboxy-benzonitril, 97% (Synthon Chemicals), diacrylate 1,4-bis [4-(6-acyloyloxyhexyloxylbenzoyloxy], 97% (Synthon Chemicals), E7 liquid crystal (Merck Millipore,), S-811 chiral dopant: S-octan-2-yl 4-((4-(hexyloxy)benzoyl)oxy) benzoate (Merck Millipore), photo-initiator: phenylbis(2,4,6-trimethylbenzoyl) phosphine oxide, 97.0% (Irgacure 819; Sigma-Aldrich). Dichloromethane HPLC grade was purchased from Sigma-Aldrich and ethanol (99.9%) was purchased from

VWR. The synthesis of the azobenzene switch is described as a Supplementary Method.

**Preparation of the flat and twisted ribbons**. Glass slides (Thermo Fischer Scientific) were cut into 25 × 25 mm squares with the help of a diamond tip pen. The glass squares were immersed in ethanol and sonicated 30 min at room temperature. The slides were air-dried and spin-coated with a Sunever 150 alignment layer polyimide (Nissan Chemical Industries) using a Laurell WS-650 spin coater. The slides were heated on a hot plate at 180 °C for an hour. Alignment was applied by rubbing the slides with a velvet cloth. Spacers of 50 μm were placed on both edges of the slides and the cell was assembled with epoxy glue. For tensile experiments, cells with unidirectional planar alignment were fabricated, while springs were obtained by using twist alignment cells.

The monomers and the photo-initiator were weighted separately, dissolved in dichloromethane and mixed in a glass vial. Dichloromethane was then evaporated by heating the vial in an oven, in the dark, at 60 °C overnight. The liquid crystal that was obtained was then heated at 80 °C for 30 min.

The glass cells were pre-warmed at 60 °C and the mixture was introduced into the cell by using a metallic tip. Once the cell was filled, it was covered by aluminum foil and heated at 80 °C for a few minutes to prevent crystallization. After cooling back the cell, a cutoff filter ($\lambda > 425$ nm, Edmund Optics) was placed on top of it and photo-polymerization was performed with an Edmund Optics MI-150 high-intensity illuminator over 3 h. The polymerization temperature was adjusted for

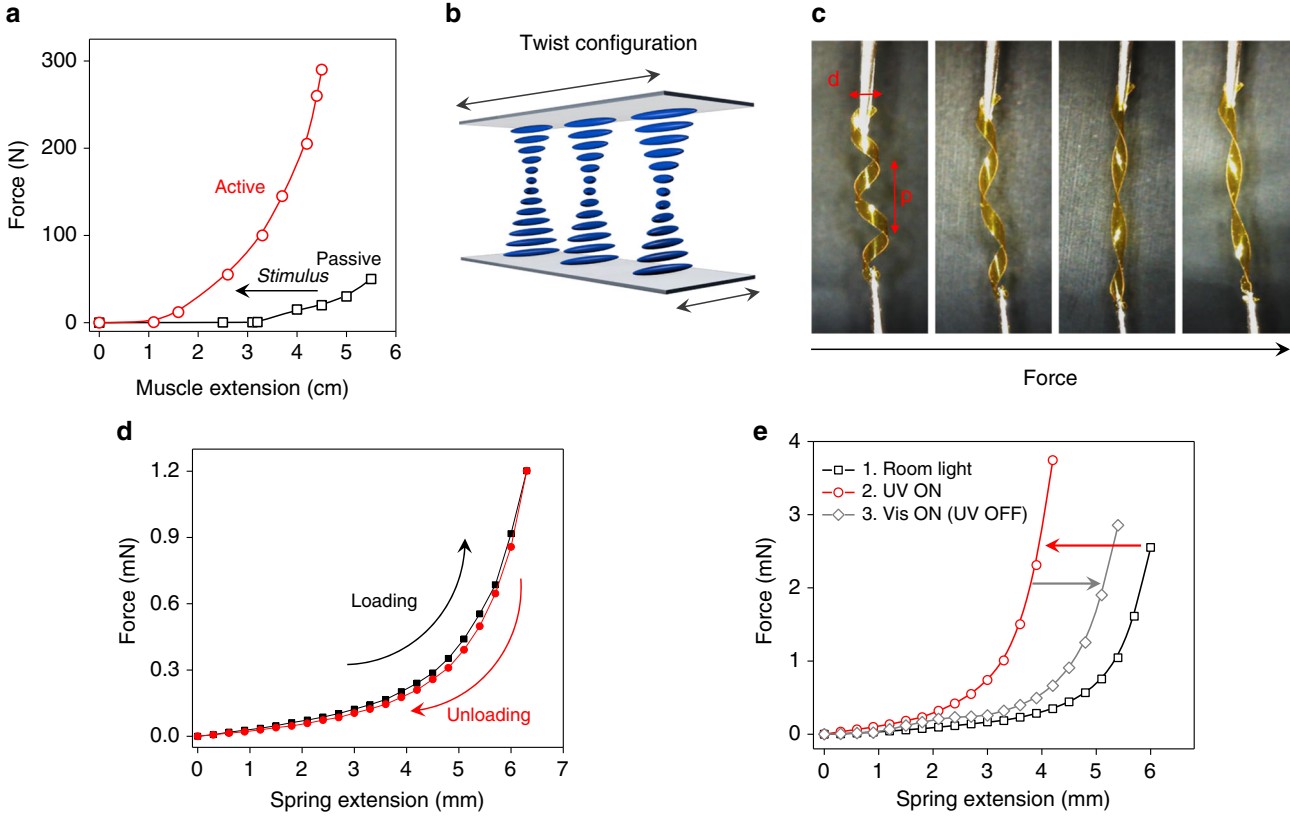

**Fig. 5** Muscle-like mechanical behavior from springs of light-stiffening materials. **a** Nonlinear extension of passive and active muscle fibers in response to a pulling force. Reproduced from R. Shadmehr & M. A. Arbib, *Biol. Cybern.* 66, 463–477 (1992). **b** Twist configuration of liquid crystals in a glass cell. **c** Response of a liquid crystal polymer spring to increasing pulling forces. **d** Loading (increasing pulling force) and unloading (decreasing pulling force) of the springs, displaying nonlinear and reversible strain stiffening, in which hysteresis is negligible. **e** Muscle-like behavior of ground state (black) and light-activated (red) polymer springs as shown from force-extension curves (here the molar ratio between the free liquid crystal and the azobenzene is $R = 4.7$)

each mixture. The cells were eventually placed in an oven at 60 °C overnight for post-curing.

**Characterization of the ribbons**. The ribbons were analyzed by attenuated total reflection infrared spectroscopy using a Nicolet-6700 FT-IR (Thermo Scientific) for traces of unreacted acrylate (Supplementary Fig. 10).

The elastic moduli were measured with a tensile tester (Zwick Roel Z1.0). Ribbons cut out of thin films were irradiated from one side with an LED lamp ($\lambda = 365$ nm), with a measured irradiation power of 17 mW/cm². For ex situ photo-tensile experiments the ribbons were irradiated for 3 min to reach the photo-stationary state, then the samples were stretched under continuous illumination. For in situ tensile experiments, the flat ribbons were initially stretched without illumination. After the establishment of a linear stress–strain curve the ribbons were illuminated and the tensile test was allowed to continue.

The force–extension curves were acquired on the springs with a home-made setup. The springs were clamped on one end with magnets, and on the other hand they were strained by a tweezer attached to a step motor. The system was placed on a balance to measure the gravitational force applied to the spring. The spring was brought back to its initial position and irradiated with an LED lamp ($\lambda = 365$ nm, 17 mW/cm²) until the photo-stationary state was reached. Then the test was repeated to obtain the force-displacement curve in the UV-activated state.

Atomic force microscopy was performed using an NTEGRA spectrometer from NT-MDT. The images were recorded in tapping mode and displayed as phase images.

## Data availability

The authors declare that the data supporting the findings of this study are available within the paper and its supplementary information files. Additional data on methods used are available from the corresponding author upon reasonable request.

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

## Acknowledgements

This research was supported financially by the Dutch Research Council (NWO Grant 13PR3105), the European Research Council (ERC Consolidator Grant 772564), and the Volkswagen Foundation (Integration of Molecular Components in Functional Macroscopic Systems 93424). We thank Tibor Kudernac for useful advice and Luca Ricciardi for help with the illustrations.

## Author contributions

N.K., F.L. and A.R. designed the experiments. F.L., A.-D.N. and A.R. performed the experiments. S.K. synthesized the molecular switches. N.K. and F.L. wrote the paper. All authors analyzed the data, edited the manuscript and discussed the results at all stages.

## Competing interests

The authors declare no competing interests.
