## [Peer Review File · Nature Communications]

Reviewers' comments:

Reviewer #1 (Remarks to the Author):

The authors have adequately responded to our and the other reviewers critiques. Further, this manuscript as written is more appropriate for Nature Communications. I recommend publication.

Reviewer #2 (Remarks to the Author):

The authors basically answer the proposed questions, and it looks much better now.

Reviewer #3 (Remarks to the Author):

This manuscript describes the change in mechanical properties of a well-known liquid crystal elastomer mixture. The photoresponsive properties in terms of shape transformations have been described in detail for this and alike materials. Now the authors focus their attention to the stiffness of the material.

In my earlier review, I called the change in moduli modest, but the authors in their reply assure me that their observed changes are 'unprecedented'. In fact, they use the word rather abundantly. Fully reversible light-induced stiffness changes of a factor 2 (stiffening or softening) have been reported for soft materials (hydrogels) before. Even a change of a factor 10 has been published. This is why I called the observed changes modest.

Since the effects are relatively small and the materials are well-known, the impact is rather limited to me. In addition, the manuscript suffers from severe (unprecedented) window dressing.

Additional comments:

Figure 4, earlier in the Suppl Mater still shows contradictions: A photostiffening material (Fig 4a) shows the highest stiffness at the highest intensities, but in Fig. 4e, the order is swapped and the sample with the lowest irradiation intensity is stiffest.

Besides the authors argue that the effect is absent with an isotropic liquid present (requested control, rejected by the authors). Looking at Fig. 4a again, the highest stiffening effects are measured in the material with the highest irradiation intensity, which is almost certainly isotropic due to heating.

For reversibility, I don't understand why the authors do not simply clamp in a straight piece of material in their tensile tester, apply a small load and apply 5 cycles of irradiation with UV and vis. This seems like a much simpler, easier to grasp experiment than the twisted springs in Figure 5. The effect is claimed to come from interfacial tension between the different phases in the material. This claim (that is the basis of the mechanism) is unsupported and not so difficult to test, I believe.

Response to Reviewer #3

We would like to thank Reviewer #3 for the time taken to review the new draft of the manuscript. Several of the comments have made us review our text and figure captions once more, and as a result of the comments by Reviewer #3 the document has gained in clarity. Please see below our responses to the comments, and our account of how we have acted on them.

This manuscript describes the change in mechanical properties of a well-known liquid crystal elastomer mixture. The photoresponsive properties in terms of shape transformations have been described in detail for this and alike materials. Now the authors focus their attention to the stiffness of the material.

Response: Please note that the mechanical properties we evidence are not those of the liquid crystal elastomer itself, but instead those of the nanostructured material based on the liquid crystal elastomer, and that the special mechanical properties arise from the interfacial properties.

#1 In my earlier review, I called the change in moduli modest, but the authors in their reply assure me that their observed changes are ‘unprecedented’. In fact, they use the word rather abundantly.

Response: We agree with reviewer #3 that we have overused the term “unprecedented”. This term no longer appears in the manuscript.

Fully reversible light-induced stiffness changes of a factor 2 (stiffening or softening) have been reported for soft materials (hydrogels) before. Even a change of a factor 10 has been published. This is why I called the observed changes modest.

Response: We thank the reviewer for this valuable information, and agree that a lot has been achieved with hydrogels before. In response, we have replaced “soft matter” with “polymers” throughout the document, whenever possible, to make our claims more specific. We have also added one reference on stimuli-induced mechanical changes in hydrogels [46], to complement the two references that we had already included in the manuscript [39,40].

As a further response (and explanation), hydrogels are indeed soft materials but their modulus is typically quite modest, which makes it easier to encode changes in elastic modulus, but represents a limitation in the production of useful macroscopic work.

We have added the following paragraph to accommodate the comments by reviewer#3:

“A comparable magnitude in softening and stiffening behavior has been observed in temperature-responsive hydrogels, by collapse of polyethylene glycol polymer chains [39,40]. In hydrogels doped with molecular motors, light induced an increase in the shear modulus of the material, that relates to viscosity [46]. However, stimuli-induced increase of the Young Modulus was not reported, and importantly the modulus of hydrogels overall remains quite modest, which constitutes a major limitation in the production of useful macroscopic work.”

#2 Since the effects are relatively small and the materials are well-known, the impact is rather limited to me.

Response: we trust that with the specifications provided above, *i.e.* that the relative modulus can increase up to 154% its initial value in a material that has a significantly higher Young Modulus than a hydrogel, Reviewer #3 will be satisfied that this effect is not small for our type of stiffer (and shape-shifting) polymer systems.

#3 In addition, the manuscript suffers from severe (unprecedented) window dressing.

Response: Without the naming of specific instances of this ‘window dressing’, it is difficult for us to respond to this comment. We have examined the entire manuscript, taken out the use of the term ‘unprecedented’ and checked all the language. We are excited by, and proud of the results. It would be only natural for this to come across in the presentation, and we will be happy to accommodate specific request from the Editor if any of this might be seen as ‘window dressing’.

#4 Figure 4, earlier in the Suppl Mater still shows contradictions: A photostiffening material (Fig 4a) shows the highest stiffness at the highest intensities, but in Fig. 4e, the order is swapped and the sample with the lowest irradiation intensity is stiffest.

Response: The curve in Fig. 4a does not relate to light-induced stiffness changes of the material, instead it is related to its light-induced shape changes, *i.e.* in other words it shows the potential for this material to convert light into mechanical work.

To avoid confusion, we have added this specific sentence to the Figure caption and we have also specified that, in contrast, Figure 4e shows how the material responds to strain by mechanical stretching. Although the readout is “stress” in both cases, the experiments reported in Fig. 4a and the one in Fig. 4e are actually very different, the former relating to the photogeneration of stress (by shape change), and the latter to stiffness changes. This should make it clear that there is no contradiction.

Besides, Reviewer #3 is correct that in Fig. 4e the material shows higher stiffness for lower intensities of irradiation. That is because, at lower irradiation intensities, the thermal effect is negligible and therefore the stiffening effect is not counteracted by the photothermal (softening) effect. When the light intensity is so high that the temperature increase is significant (see Fig. 4b), the stiffening is still occurring however its effect is mitigated by the heating effect. We have also changed a few sentences in the paragraph on illumination intensity (Page 7), in order to enhance the clarity of the paper.

#5 Besides the authors argue that the effect is absent with an isotropic liquid present (requested control, rejected by the authors).

Response: Unfortunately, the experiment proposed by the reviewer is not feasible. This is because the material must be prepared with a relatively large component of isotropic liquid, and the presence of this isotropic liquid compromises the liquid crystalline character of the material - consequently that experiment would not provide any valuable information.

An alternative would be to prepare the material with a non-polymerizable liquid crystal, remove the liquid crystal, and replace it by an isotropic liquid at a later stage. We actually tried this, but we found out that after removal of the liquid crystal by acetonitrile, the network does not swell significantly in a range of isotropic solvents.

For clarity, we have added a paragraph in the manuscript:

“We note that it was not possible to prepare a similar material where the pores would be filled with an isotropic liquid, because the presence of an isotropic liquid in large quantities compromises the liquid crystallinity of the polymer network.”

#6 Looking at Fig. 4a again, the highest stiffening effects are measured in the material with the highest irradiation intensity, which is almost certainly isotropic due to heating.

Response: With regard to Figure 4a, as mentioned above, this experiment does not relate to stiffness changes, it shows how much stress can be generated by the shape-shifting material. We fully accept that this was confusing and we have stated this now clearly in the Figure caption.

#7 For reversibility, I don't understand why the authors do not simply clamp in a straight piece of material in their tensile tester, apply a small load and apply 5 cycles of irradiation with UV and vis. This seems like a much simpler, easier to grasp experiment than the twisted springs in Figure 5.

Response: We show reversibility for a “straight piece of material” in Supporting Figure S5. However, in the main text we chose to show reversibility in the springs, because the mechanical behavior shows both strain-stiffening and photo-stiffening, and their non-linear mechanical behavior is thus closer to that of muscle fibers, as shown in Figure 5a.

#8 The effect is claimed to come from interfacial tension between the different phases in the material. This claim (that is the basis of the mechanism) is unsupported and not so difficult to test, I believe.

Response: Our evidence is that contact angle experiments (Figure 3d) show an increase in interfacial tension between free liquid crystal and the polymer network, as a response to light. In addition to this experimental proof, we also support our conclusions with previous works reporting an increase in interfacial tension upon illumination (Ref. 48 and Ref. 49).

REVIEWERS' COMMENTS:

Reviewer #3 (Remarks to the Author):

The authors have addressed my major concerns. It is still a bit disturbing that they prefer to measure fairly straightforward properties in a complex geometry (measure a change in the material stiffness in helical strip with numerous potential artifacts). In addition, the reversibility experiment is still missing (not as the authors claim in the SI).

Response to Reviewer #3

The authors have addressed my major concerns. It is still a bit disturbing that they prefer to measure fairly straightforward properties in a complex geometry (measure a change in the material stiffness in helical strip with numerous potential artifacts).

Our response: Except for the experiments shown Figure 5, all measurements reported in this manuscript have been performed on a flat piece of material, as can be seen from the scheme in Figure 3a. In other words, the experiments shown in Figure 3 and in Figure 4 are all measured in a simple, flat geometry.

We now refer explicitly to the flat geometry of the ribbons, whenever possible.

In addition, the reversibility experiment is still missing (not as the authors claim in the SI).

Our response: We maintain that the reversibility of the light-induced stiffening is shown in Supplementary Figure 5. The graph shows that when the molecular switches convert back from the *cis* form to the *trans* form, the slope of the stress-strain curve is lower, which means that the Young Modulus decreases back.

We write : “*Reversibility in the photo-stiffening behavior is shown in Supplementary Figure 5, where irradiation of a flat ribbon with visible light converts the azobenzene back to its trans form, and the Young modulus consequently returns back to a value that is comparable to its original value.*”